# Ticagrelor Resistance in Cardiovascular Disease and Ischemic Stroke

**DOI:** 10.3390/jcm12031149

**Published:** 2023-02-01

**Authors:** Song He, Yapeng Lin, Quandan Tan, Fengkai Mao, Kejie Chen, Junli Hao, Weidong Le, Jie Yang

**Affiliations:** 1Department of Neurology, The First Affiliated Hospital of Chengdu Medical College, Chengdu 610072, China; 2International Clinical Research Center, Chengdu Medical College, Chengdu 610072, China; 3School of Public Health, Chengdu Medical College, Chengdu 610072, China; 4School of Biomedical Sciences and Technology, Chengdu Medical College, Chengdu 610072, China; 5Institute of Neurology, Sichuan Provincial People’s Hospital, University of Electronic Science and Technology of China, Chengdu 610072, China; 6Department of Neurology, Sichuan Provincial People’s Hospital, University of Electronic Science and Technology of China, Chengdu 610072, China; 7Sichuan Provincial Key Laboratory for Human Disease Gene Study, Chengdu 610072, China

**Keywords:** cardiovascular disease, ischemic stroke, ticagrelor resistance, high on-treatment platelet reactivity, genetics, epigenetics

## Abstract

Ticagrelor, acting as a reversible platelet aggregation inhibitor of P2Y12 receptors (P2Y12R), is regarded as one of the first-line antiplatelet drugs for acute cardiovascular diseases. Though the probability of ticagrelor resistance is much lower than that of clopidogrel, there have been recent reports of ticagrelor resistance. In this review, we summarized the clinical application of ticagrelor and then presented the criteria and current status of ticagrelor resistance. We further discussed the potential mechanisms for ticagrelor resistance in terms of drug absorption, metabolism, and receptor action. In conclusion, the incidences of ticagrelor resistance fluctuated between 0 and 20%, and possible mechanisms mainly arose from its absorption and receptor action. Specifically, a variety of factors, such as the drug form of ticagrelor, gut microecology, and the expression and function of P-glycoprotein (P-gp) and P2Y12R, have been shown to be associated with ticagrelor resistance. The exact mechanisms of ticagrelor resistance warrant further exploration, which may contribute to the diagnosis and treatment of ticagrelor resistance.

## 1. Introduction

Cardiovascular disease (CVD) and ischemic stroke (IS) are the leading causes of disability and death in the world [1,2]. Antiplatelet agents have become the cornerstone in the treatment and prevention of CVD and IS. Dual antiplatelet therapy (DAPT), consisting of aspirin and a P2Y12R antagonist, is commonly acknowledged as a vital approach in treating patients with CVD or acute minor IS, especially after endovascular treatment [3,4,5].

Clopidogrel is currently regarded as the traditional and most used oral P2Y12R inhibitor. Active metabolites of clopidogrel bind with P2Y12R, irreversibly causing the receptor to be permanently inactive. However, the incidence of clopidogrel resistance is approximately 30% [6]. Being different to clopidogrel, ticagrelor is an oral antiplatelet drug, whose prodrug could function directly, quickly, and reversibly, without being activated by the metabolism of hepatic enzymes. Hence, ticagrelor resistance is significantly lower than clopidogrel resistance. Ticagrelor has become one of the first-line antiplatelets for acute CVD and is now widely used as an alternative to clopidogrel for IS [3,4,7].

High on-treatment platelet reactivity (HTPR) is regarded as a useful alternative standard for antiplatelet resistance [8]. Although ticagrelor has a lower possibility of resistance than clopidogrel, it has been noticed that ticagrelor resistance is not rare [7]. In these ticagrelor-resistant patients, the occurrence of ischemic thrombotic events is more frequent than that of ticagrelor responders [9]. Therefore, ticagrelor resistance is a critical issue as there are currently few alternative drugs. 

In summary, it is of great importance to evaluate the possible mechanisms and find potential solutions for ticagrelor resistance. In this review, we searched MEDLINE (via PubMed) and Embase (via OVID) from their inception to 1 October 2022 for the relevant articles written in English, using the main terms “ticagrelor”, “resistance “, “high on-treatment platelet reactivity”, “cardiovascular disease”, and “ischemic stroke”. Then, we summarized the clinical application of ticagrelor and the research status of ticagrelor resistance, and we further discussed the possible mechanisms and potential solutions for ticagrelor resistance.

## 2. Clinical Application of Ticagrelor

Antiplatelet agent therapy is the cornerstone of pharmacological treatment for preventing both CVD and IS. The following is a brief summary of the clinical applications of ticagrelor for CVD and IS.

### 2.1. Application of Ticagrelor for CVD

Due to its rapid onset of action and more effective and lower ratio of resistance, many international guidelines currently recommend ticagrelor as the first-line choice for the treatment of dual antiplatelets for acute CVD, especially for those with percutaneous coronary intervention (PCI).

The 2016 AHA guideline recommended the use of ticagrelor, and it is preferred over clopidogrel in patients with NSTE-ACS or STEMI treated with DAPT after PCI (II, a) [10]. The 2021 ESC guidelines recommended DAPT with ticagrelor or prasugrel as the standard antiplatelet therapy for ACS patients and for patients with chronic coronary syndromes (CCS) undergoing elective PCI, especially after complex interventions (I, a) [11]. As for the choice between prasugrel and ticagrelor, there are still conflicting data [12,13,14].

### 2.2. Application of Ticagrelor for IS

According to the current guidelines for the antiplatelet treatment for acute ischemic stroke (AIS) and high-risk transient ischemic attack (TIA), short-time DAPT with clopidogrel and aspirin are still the most recommended drugs (I, a) [15].

Although ticagrelor is superior to clopidogrel for acute CVD, its superiority for AIS and TIA presently remains controversial (Table 1).

When comparing ticagrelor to aspirin, according to a report based on the SOCRATES trial, ticagrelor was not superior to aspirin in treating patients with non-severe AIS or high-risk TIA who were not considered to have had a cardioembolic stroke [16,17].

Determining whether ticagrelor is superior to clopidogrel in DAPT with an aspirin background is still controversial. Some studies have suggested that there is no statistical difference in the efficiency and safety of ticagrelor and clopidogrel [18]. However, there are also studies that have demonstrated that in stroke patients, the antiplatelet effect of ticagrelor is superior to that of clopidogrel [7,19,20]. For example, the PRINCE trial showed that patients in a ticagrelor plus aspirin group had a lower proportion of HTPR than those in a clopidogrel plus aspirin group, particularly for those who were carriers of the CYP2C19 loss-of-function allele [19]. Recently, the CHANCE 2 trial drew a similar conclusion wherein the risk of a 90-day stroke with a ticagrelor plus aspirin group was slightly lower than that of a clopidogrel plus aspirin in Chinese patients with mild AIS or TIA and who carry the CYP2C19 inactivated allele [7].

In testing whether ticagrelor plus aspirin is superior to aspirin alone in patients with IS or TIA, the THALES trial found that the combination of ticagrelor and aspirin reduced the rate of recurrent stroke and death at the cost of more bleeding events by 30 days [5].

In summary, ticagrelor is a faster acting and more potent antiplatelet, and it should be considered as one of the first-line antiplatelet drugs for acute CVD and as an alternative strategy for IS patients who carry the CYP2C19 inactivated allele. However, whether ticagrelor alone or in combination with aspirin should be considered as the optimal therapy of general AIS is still under investigation.

**Table 1 jcm-12-01149-t001:** Summary of clinical trials that compare the efficiency of ticagrelor with others for IS.

Study	Subjects	Experimental/Control	Conclusions
SOCRATES (2017) [16]	Patients with AIS or TIA	Ticagrelor/aspirin	Ticagrelor monotherapy was not significantly better than aspirin monotherapy (both initiated within 24 h of symptom onset).
PRINCE (2019) [19]	Patients with acute mild stroke or TIA	Ticagrelor plus aspirin/clopidogrel plus aspirin	Patients in the ticagrelor plus aspirin group had a lower proportion of high platelet reactivity than those in the clopidogrel plus aspirin group, particularly those who were carriers of the CYP2C19 loss-of-function allele.
CHANCE 2 (2021) [7]	patients with mild ischemic stroke or TIA	Ticagrelor plus aspirin/clopidogrel plus aspirin	The risk of a 90-day stroke after ticagrelor was slightly lower than that of clopidogrel, and there was no difference in the risk of severe or moderate bleeding between the two groups, but the total number of bleeding events in the ticagrelor group exceeded that of the clopidogrel group.
TALES (2020) [5]	Patients with AIS or TIA	Ticagrelor plus aspirin/aspirin	The combination of ticagrelor and aspirin reduced the possibility of the primary outcome of stroke and death at the cost of more bleeding events compared with aspirin alone by 30 days.

## 3. Current Status of Ticagrelor Resistance

### 3.1. Criteria for Ticagrelor Resistance

#### 3.1.1. Methods and Instruments for Evaluating Ticagrelor Resistance

Platelet function tests are useful to evaluate ticagrelor resistance. However, the standard methods for platelet function testing are not currently uniform.

Among the common platelet function tests (Table 2 and Figure 1), light transmission aggregometry (LTA) is considered the gold standard [21]. Although accurate, LTA has its limitations. Firstly, LTA cannot assess whole-blood directly. Secondly, it is time-consuming. Thirdly, there are many variables in an operational session that can lead to differences in results due to manual operation. Inaccurate sample concentration and different operators may cause variations in the results [22,23]. 

The vasodilator-stimulated phosphoprotein-phosphorylation (VASP-P) assay is another assay specially used to detect the inhibition effect of P2Y12R [24]. The assay has the advantage of specifically reflecting P2Y12R inhibition without being affected by other receptor antagonists. However, vasodilator-stimulated phosphoprotein detected by the assay does not directly reflect platelet aggregation function. In addition, similar to LTA, VASP-P is fully manual and time-consuming, which has prevented its widespread clinical application.

For both being whole-blood assays, time-saving, and highly standardized in terms of their operation, VerifyNow, PFA P2Y, and Multiplate are three instruments that are currently widely used as alternatives to LTA to detect P2Y12R inhibitor resistance in clinical practice [25,26,27,28,29,30].

Because of their different detection principles, there are slight differences in their use scenarios.

The VerifyNow assay is a user-friendly, turbidimetric-based, agonist-induced, and cartridge-based method that measure the platelet aggregation taking place on fibrinogen-coated beads loaded in the standard cartridges. 

Similar to VerifyNow, the PFA P2Y assay is another user-friendly, agonist-induced, cartridge-based method designed to measure platelet function by detecting the time for blood to block a membrane coated with collagen and ADP. As for the comparison of PFA P2Y and VerifyNow for the detection of P2Y12R inhibitors resistance, there are reports that suggest that the sensitivity of PFA P2Y in assessing clopidogrel resistance is comparable to that of VerifyNow P2Y12 assay [27], but there has been limited experience with detecting ticagrelor resistance till now.

Multiplate is a multiple electrode impedance aggregometer that monitors the platelet function by detecting the platelet aggregation on the surface of two pairs of electrodes. As for the comparison of Multiplate and VerifyNow, the latter is more automated and demonstrates a stronger relationship to major adverse cardiovascular events (MACE) in patients with acute coronary syndromes [31]. Thus, in current clinical practice, VerifyNow is more recommended, because it is fully automated, time-saving, and has reproducible results (Figure 1) [6]. 

**Table 2 jcm-12-01149-t002:** Summary of the basic characteristics of the common platelet function assays.

Assays	Suit for	Mechanism	Definition Cut-Offs	Advantages	Disadvantages
LTA	All situations involving platelet aggregation	Detecting the optical density decrease after stimulation of the platelets.	Controversial (the reported optimal cut-offs ranged from 46–57% maximal aggregation [28,32]) ≥the cut-off: ticagrelor-resistant <the cut-off: ticagrelor-sensitive	Gold standard to measure platelet function.	Cumbersome steps, difficult to prepare samples of platelet rich plasma, trained operators required, and time-consuming.
VASP-P	P2Y12R inhibitors	Whether P2Y12R is inhibited regulates the level of VASP-P.	Controversial (the reported optimal cut-offs range from 48–61% PRI * [24]) ≥the cut-off: ticagrelor-resistant <the cut-off: ticagrelor-sensitive	Specifically reflecting the inhibition of P2Y12R pathway, whole-blood samples.	Indirect test, cumbersome steps, trained operators required, and time-consuming.
VerifyNow	COX-1 inhibitors, P2Y12R inhibitors	Detecting the aggregation of fibrinogen-coated polystyrene beads mediated by platelets.	≥208 PRU *: ticagrelor-resistant <208 PRU: ticagrelor-sensitive [33]	User-friendly (standard cartridge), fast (5min), fully automated, reproducible, and whole-blood samples.	Expensive.
PFA P2Y	COX-1inhibitors, P2Y12R inhibitors	Detecting the time for blood to block a membrane coated with collagen and epinephrine or ADP.	CT * < 106 s: ticagrelor resistant CT ≥ 106 s: ticagrelor sensitive [25,26,27]	User-friendly (standard cartridge), fast (5 min), whole-blood samples.	Limited experience with detecting ticagrelor resistance.
Multiplate	COX-1 inhibitors, P2Y12R inhibitors	Detecting the platelet aggregation on the surface of two pairs of electrodes.	≥468 AU *: ticagrelor resistant <468 AU: ticagrelor sensitive [33]	User-friendly, reproducible, and whole-blood samples.	Semi-automated.

* PRI, platelet reactivity index; PRU, platelet reactivity units; CT, closure time; and AU, aggregation units.

#### 3.1.2. Thresholds for Defining Ticagrelor Resistance

Besides the choice of methods and instruments, another critical issue is the threshold used to define ticagrelor resistance or HTPR.

Currently, most definitions of HTPR are based on the “absolute value” of the platelet aggregation function. For example, the recommended cut-offs for this definition are 208 PRU with VerifyNow, 106 s with PFA PLA, and 46 AU with Multiplate (Figure 1, Table 2) [34,35]. Patients with platelet aggregation function values above the cut-off values are defined as ticagrelor-resistant or as having HTPR. 

Instead of the “absolute value” definition, there are also a few studies that used the “relative ratio” definition, which focuses on the decreased rate of the platelet aggregation function before and after antiplatelet therapy. In these studies, the target decrease rate was defined as 30–90%, namely, patients with a PRU value that decreased less than 30% from the baseline were defined as having ticagrelor resistance or HTPR [18,36]. Though the “relative ratio” definition has the potential to provide more meaningful information than the “absolute value” definitions of true ticagrelor resistance [37], there are difficulties with carrying out the “relative ratio” definition for repeating detection in clinical practice.

In summary, the VerifyNow PFA, and the Multiplate assay using “absolute value” definitions are now more commonly used in clinical practice to define ticagrelor resistance or HTPR.

### 3.2. Current Status of Ticagrelor Resistance

To the best of our knowledge, ticagrelor resistance is less common than clopidogrel resistance, but it is not rare. With the increasing applications for ticagrelor, some reports have recently emerged with respect to ticagrelor resistance. 

Here, we reviewed related clinical trials. We found that the rate of ticagrelor resistance fluctuated between 0 and 20%. Differences in testing time points, methods used to define the resistance, ethnicity of the population, and sample size may be responsible for the fluctuation in the rates of ticagrelor resistance (Table 3) [19,20,38,39]. Ticagrelor resistance is most commonly seen in elderly patients with comorbidities, such as diabetes and obesity [18,40,41].

## 4. Pharmacokinetics and Pharmacodynamics of Ticagrelor

Before discussing the possible mechanisms of ticagrelor resistance, we briefly summarized the pharmacokinetic and pharmacodynamics profile of ticagrelor from absorption, metabolism, and action.

### 4.1. Absorption

Ticagrelor is absorbed into the blood stream through the intestinal epithelial cells. The absorbing speed of ticagrelor is obviously faster than other commonly used P2Y12R inhibitors, such as clopidogrel [42]. As a result, ticagrelor is used more often to induce immediate platelet inhibition 2–3 days before PCI or digital subtraction angiography (DSA).

### 4.2. Metabolism and Action

After being absorbed into the blood, as a non-thienopyridine, ticagrelor can directly bind with P2Y12R on the platelets and competitively inhibit the adenosine diphosphate (ADP) binding to P2Y12R, thereby inhibiting platelet aggregation [43]. Ticagrelor can also be metabolized by the liver to produce active metabolites. Cytochrome P450 3A4 (CYP3A4) and CYP3A5 are the major enzymes that transform the ticagrelor prodrug to active metabolites in the liver (Figure 2) [44]. The major active metabolite of ticagrelor in blood plasma is called AR-C124910XX, and it has almost the same antiplatelet activity when compared to the ticagrelor prodrug [45,46,47]. Therefore, different to clopidogrel, ticagrelor can exert good antiplatelet effects without relying on hepatic enzymes for its metabolism, and it has a faster onset of action than clopidogrel with a lower resistance rate [48].

## 5. Possible Mechanisms and Solutions for Ticagrelor Resistance

### 5.1. Absorption

One important factor regulating the absorption of ticagrelor is the expression of P-glycoprotein (P-gp) on the intestinal epithelia. Ticagrelor has been shown to be a substrate of P-gp, an energy-dependent membrane transporter that belongs to the ABC superfamily, which is also known as multidrug resistance protein 1 (MDR1) or ATP-binding cassette sub-family B member 1 (ABCB1). P-gp can transport a wide range of drugs and various compounds with different structures out of cells at many biological interfaces, such as the brush marginal membrane of the intestine, thus reducing the transmembrane absorption and oral bioavailability of oral drugs (Figure 2). In this way, P-gp plays a crucial role in the resistance mechanisms of many drugs. Recent studies have shed light on the expression of P-gp being an important factor that regulates the intestinal absorption of ticagrelor, and it correlates closely with ticagrelor resistance [49,50,51,52]. The expression of P-gp is influenced by many factors. First of all, genetic polymorphisms of the ABCB1 (MDR1) gene may be one crucial factor in them. There have been many studies observing the associations between lower P-gp expression and the single nucleotide polymorphism concerning 3435C > T, 1236C > T, 2677G > T/A, as well as 3435C > T [53,54,55]. However, a recent randomized controlled trial (RCT) obtained a different result that the genetic polymorphisms of ABCB1 did not differ between ticagrelor-resistant patients and ticagrelor sensitive patients [56]. Secondly, the role of the gut microbiome has been a hot topic in recent years, and recent studies have shed light on how the gut microbiome plays a significant role in regulating the expression of P-gp. Sage et al. found that specific genera of microbiome (Clostridia and Bacilli classes) are necessary for P-gp induction on the intestinal epithelia in mouse models. Metagenomic analysis of these core microbial communities revealed that some specific short-chain fatty acids and secondary bile acid production such as lithocholic acid (LCA), deoxycholic acid (DCA), and ursodeoxycholic acid (UDCA), are positively associated with P-gp expression (Figure 2) [57,58]. These results suggested that high expressions of these metabolites of gut microbiome may be associated with high P-gp expression, which in turn mediates the efflux transport of ticagrelor and results in ticagrelor resistance.

Besides these, the form of ticagrelor may also be an important factor influencing its absorption. Some recent studies have brought to light that different compositions of ticagrelor may contribute to improving its bioavailability. To compare the bioavailability and safety of the ticagrelor powder suspended in water with that of whole tablets administered orally, Teng et al. conducted an RCT experiment with healthy volunteers and reached a conclusion that ticagrelor in crushed powder form results in increased plasma concentrations of ticagrelor [59]. In addition, there have been reports suggesting that ticagrelor powder alleviates the resistance of ticagrelor in cardiovascular patients [18]. Further studies are warranted to examine the efficacy of crushed powder over whole-tablet administration for ticagrelor resistance.

### 5.2. Metabolism

As we mentioned in 4.2, CYP3A4 and CYP3A5 are the major metabolizing enzymes that transform the ticagrelor prodrug to active metabolites. However, the ticagrelor prodrug has the same functions as active metabolites, and there has been no evidence proving a direct connection between CYP3A4/CYP3A5 polymorphisms and ticagrelor’s effects until now [44,60].

### 5.3. Action

Ticagrelor exerts an anti-platelet aggregation effect through the competitive inhibition of P2Y12R on platelet membranes. Therefore, the expression and function of P2Y12R are closely related to the effects of ticagrelor.

As platelets are anucleated and without DNA, epigenetic differences become the most likely factors to be involved in the regulation of P2Y12R in platelets. Studies focusing on microRNA (miRNA) have made breakthroughs in exploring the correlations between specific miRNAs and the expression of P2Y12R in platelets (Figure 2) [61,62,63,64]. Researchers have observed that the levels of some miRNAs (miRNA-223-3p, miRNA-126, miRNA-365-3p, and miRNA-339-3p, et al.) are positively correlated with the incidence of HTPR [63,64]. Among these miRNAs, Patricia et al. further suggested that the expression of P2Y12R may be controlled by miRNA-223 in human platelets [62].

Although there remains a lack of high-level evidence about whether these epigenetic factors affecting P2Y12R have a direct impact on the antiplatelet effects of ticagrelor, some studies have indicated that there is an association between specific miRNAs and platelet reactivity after ticagrelor treatment. In patients treated with DAPT with ticagrelor plus aspirin, Chyrchel et al. observed that increased miR-223 in plasma was significantly correlated with decreased ADP-induced platelet reactivity [65]. In addition, Chen et al. observed that the expression levels of miRNA 365-3p were correlated with HTPR after ticagrelor therapy [64]. These results indicated the possibility of using specific miRNAs as biomarkers for ticagrelor resistance in the future.

## 6. Discussion and Perspective

Although ticagrelor has a lower incidence of resistance than clopidogrel, once resistance occurs, management will be a difficult issue due to few effective alternative drugs. Therefore, how to detect and deal with ticagrelor resistance is a significant topic for further exploration.

At present, the most widely used method to evaluate platelet reactivity is VerifyNow, but its testing costs are high, and the thresholds used to define ticagrelor resistance lack uniform standards. There is an urgent need to find a rapid, economical, and practical method to detect and define ticagrelor resistance.

Once ticagrelor resistance has been detected, stopping and trying alternative P2Y12R inhibitors can be one of the solutions. Prasugrel and cangrelor are two P2Y12R inhibitors to consider. In the 2021 ESC guides, prasugrel and ticagrelor were equally recommended for patients with ACS [11]. However, the differences in safety and efficacy between prasugrel and ticagrelor remain controversial [12,13,14]. Similar to ticagrelor, cangrelor is an IV, direct-acting P2Y12R inhibitor. What is different is that cangrelor is an intravenous medication, which gives it a faster onset and offset of action [66], but this also limits its application scenarios.

The mechanisms and possible solutions of ticagrelor resistance are under investigation (Figure 3). The absorption and action processes of ticagrelor have been shown to be crucial in ticagrelor resistance. First, the form of ticagrelor may be an important factor influencing the absorption and bioavailability of ticagrelor. Thus, increasing its bioavailability by changing the ticagrelor form may be a possible way to overcome ticagrelor resistance. A recent report observed that transferring a ticagrelor tablet into powder form could improve absorption and alleviate ticagrelor resistance in IS patients [18]. However, whether the result is generally effective for IS patients still requires a prospective study, or RCT. Second, P-gp is another crucial factor causing ticagrelor resistance by regulating intestinal absorption. Factors involved in the regulation of the expression of P-gp, such as genetic polymorphisms and the micro-environment of the gut, may lead to high P-gp expression, resulting in reduced ticagrelor utilization and ticagrelor resistance. However, there are still some controversial results regarding these factors, which have to be verified in further animal and cellular experiments. Third, with respect to the receptor action, epigenetic differences, especially miRNA regulating the expression of P2Y12R have been proven to be associated with the level of platelet reactivity, but the exact mechanism and affirmative effect are unclear. In addition to miRNA, other non-coding RNAs (such as lncRNA and circRNA) may also cause ticagrelor resistance by regulating the expression of P2Y12R on platelets, which needs to be confirmed by further exploration.

To sum up, the absorption and action processes are very prospective parts for future clinical application and could be considered important entry points for early monitoring and addressing ticagrelor resistance. By comparing the genetic and epigenetic differences related to these two key proteins (P-gp and P2Y12R) between resistant and sensitive patients or using animal models, we may identify new biomarkers for the early diagnosis of ticagrelor resistance. Perhaps in the future, ticagrelor resistance could be addressed by means such as intervening at relevant genetic and epigenetic polymorphisms, changing drug form, and modulating the composition of the gut microbiome.

## 7. Conclusions

With widespread clinical use, it has been noticed that the incidence of ticagrelor resistance is not rare, and its possible mechanisms primarily arise from its absorption and receptor action. Specially, a variety of factors, such as the drug form of ticagrelor, the microecology of the gut, and the expression and function of P-glycoprotein (P-gp) and P2Y12R, have been shown to be associated with ticagrelor resistance. The exact mechanisms of ticagrelor resistance warrant further exploration, and they may contribute to the early diagnosis and treatment of ticagrelor resistance in the future.

## Figures and Tables

**Figure 1 jcm-12-01149-f001:**
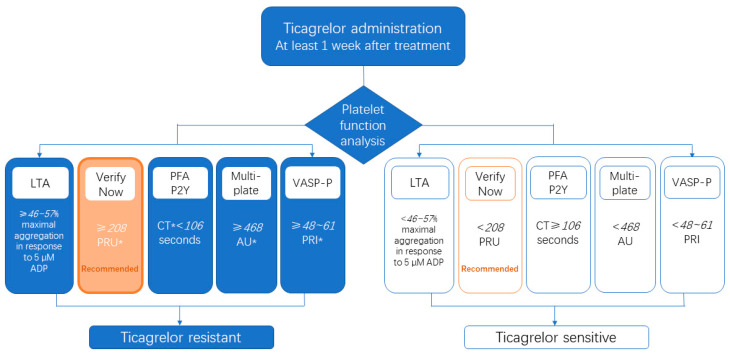
Flow chart for the diagnosis of ticagrelor resistance.* PRI, platelet reactivity index; PRU, platelet reactivity units; CT, closure time; and AU, aggregation units.

**Figure 2 jcm-12-01149-f002:**
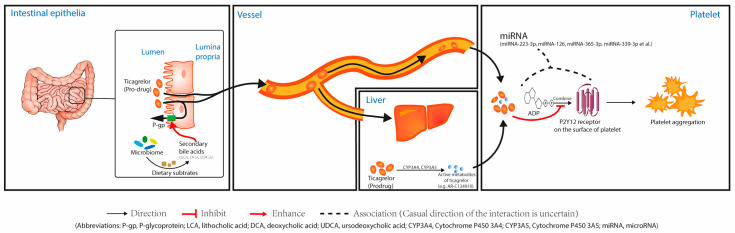
Pharmacokinetics and pharmacodynamics of ticagrelor and the factors impacting the pathways.

**Figure 3 jcm-12-01149-f003:**
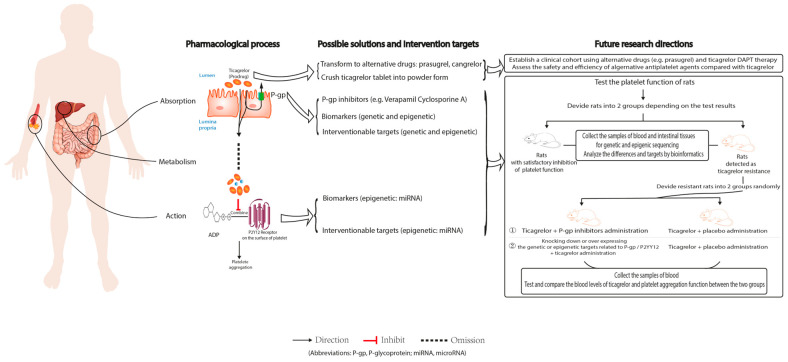
Possible intervention targets of, and future research directions for, ticagrelor resistance.

**Table 3 jcm-12-01149-t003:** Incidence of ticagrelor resistance in clinical trials.

Study	Design	Drug and Dose	Sample Size	Method and Definition	Detection Timing	Incidence of Ticagrelor Resistance (HTPR *)
Wang Y. et al. (2019) [19]	RCT	Aspirin (100 mg qd *) plus ticagrelor (180 mg loading dose on day 1, followed by 90 mg bid until day 90).	*n* = 280	VerifyNow; HTPR was defined as ≥208 PRU.	3 months after treatment.	13%
Yang Y. et al. (2020) [20]	RCT	Aspirin (100 mg qd *) plus ticagrelor (180 mg loading dose on day 1, followed by 90 mg bid until day 90).	*n* = 197	Aggrestar (PL) platelet function analyzer; HTPR was defined as MAR ≥ 35%.	3 months after treatment.	20%
Yue W. et al. (2021) [38]	Prospective cohort	Aspirin (100 mg qd) plus ticagrelor (90 mg bid).	*n* = 446	Optical heterometric method; HTPR was defined as aplatelet aggregation rate of ≥ 46%.	1 month before and after Treatment.	10%
Choi WG. et al. (2021) [39]	RCT	Aspirin plus ① ticagrelor (45mg bid) and ② ticagrelor (90 mg bid).	① *n* = 22 ② *n* = 19	VerifyNow; HTPR was defined as ≥ 208 PRU.	1 week after treatment.	0%

*: HTPR, high on-treatment platelet reactivity; qd, once daily; bid, twice daily, RCT, randomized controlled trial; and MAR, maximum aggregation ratio.

## Data Availability

Data are available on request to the Corresponding author.

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
