# Peer review of "Ticagrelor Resistance in Cardiovascular Disease and Ischemic Stroke"

_jcm, 2023, doi:10.3390/jcm12031149_

Round 1
Reviewer 1 Report
This is a useful review on ticagrelor resistence, its diffusion and its mechanisms. The topic is relevant and conclusions are supported by the evidence reviewed.
Unfortunately, I found it difficult to read most of the paragraphs, as there are many errors in the use of english language. Therefore, I think that the text needs a thorough revision of english language.
Moreover, I suggest to summarize some paragraphs, expecially the part on pharmacokinetics.
Reviewer 2 Report
Dear Authors,
Thank you for the invitation to review this paper exploring Ticagrelor resistance.
The paper is missing crucial methodology details. There are no information how the scientific literature was explored.
Ticagrelor is mentionned as "new" drug but it is in fact not a new one.
The manuscript should be improved by :
- the details of the action of ticagrelor on the platelet function.
- to insist on the clinical indication of ticagrelor as first line in patient with ACS (STEMi, NSTEMI, unstable angine), as prasugrel.
- in the tests section, no information about the PFA test with P2Y cartridge and the VASP test.
- the cut-of for LTA is not correct (same units as Verifynow ?).
- you mentionned Figure 1 and 2 in your manuscript but they are not present.
- the mention and the position of cangrelor
- a flowchart of the use of ticagrelor and the place of Resistance assesment
Overall, the whole paper needs an upgrade in the quantitative and qualitative English language. The manuscript should be reviewed by an english talking person.
A paper has just been accepted in the same field (Blaško P, Samoš M, Bolek T, Stančiaková L, Škorňová I, Péč MJ, Jurica J, Staško J, Mokáň M. Resistance on the Latest Oral and Intravenous P2Y12 ADP Receptor Blockers in Patients with Acute Coronary Syndromes: Fact or Myth? J Clin Med. 2022 Dec 4;11(23):7211. doi: 10.3390/jcm11237211. PMID: 36498785; PMCID: PMC9737839.)
Round 2
Reviewer 2 Report
Dear Authors,
Thank you for this improved version, both in terms of english and content.
Two remarks :
PFA is not the same automate as Verify now. Could you add this automate with the use of P2Y cartridge in your list of avalilable device to monitor platelet resistance.
And so adapt the Figure 1
Thank you
